# ACD: Antimicrobial chemotherapeutics database

**Mohd W. Azam, Amit Kumar, Asad U. Khan**id *

Medical Microbiology and Molecular Biology Lab, Interdisciplinary Biotechnology Unit, Aligarh Muslim University, Aligarh, India

* asad.k@rediffmail.com

**Data Availability Statement:** All relevant data are within the paper.

**Funding:** The author(s) received no specific funding for this work.

## Abstract

Antimicrobial resistance is becoming a growing health problem, which has become a challenge for the physicians to control infection and also an economic burden on the healthcare. This increase in resistance to the present antimicrobial agents led the researchers to find some alternative and more efficient drugs which can fight with the resistant microorganisms more effectively. Hence, *in silico* approach is used to design some novel drugs against various targets of microorganisms. For effective virtual screening of the drugs, there is a need to know about the chemical structure and properties of the antimicrobial agents. Therefore, we have prepared a comprehensive database as a platform for the researcher to search for possible lead molecules. Antimicrobial chemotherapeutics database (ACD) is comprised of ~4100 synthetic antimicrobial compounds as well as ~1030 active antimicrobial peptides. The Antimicrobial peptides are mainly from biological sources but some of them are synthetic in nature. Only those compounds, which are found to be active against either bacteria (both Gram-positive and negative) or fungus, are selected for this database. The ACD database is freely available at URL: http://amdr.amu.ac.in/acd, and it is compatible with desktops, smartphones, and tablets.

## Introduction

The ineffectiveness of antimicrobial agents, i.e. antimicrobial resistance (AMR), has become a global public health threat. A large number of bacterial species have been reported to have resistant markers against different antibiotics. *Klebsiella pneumoniae*, *E. coli*, *Staphlylococcus aureus*, *Mycobacterium*, *Streptococcus pneumoniae*, *etc.*, are some of the common species involved in infections [1–3]. It is a great menace to the patients having organ transplantation, major surgeries like hip transplant, breast biopsy, caesarean sections and cancer chemotherapies where the risk of spreading infection is high. Besides this, antimicrobial resistance (AMR) causes prolonged treatment of the disease and an enhanced cost of treatment [4]. As per WHO MDR-TB factsheet 2018 report about 558,000 new cases of multi-drug resistance tuberculosis (MDR-TB) have been estimated in 2017 and 161,000 cases of MDR or RR-TB have been detected and reported in 2017 [5]. The MRSA (Methicillin-resistant *Staphylococcus aureus*)

**Competing interests:** I have read the journal's policy and the authors of this manuscript have the following competing interests: [Asad U khan is editor of Plos One]" a. "This does not alter our adherence to PLOS ONE policies on sharing data and materials. b. I on behalf of all authors and my own behalf declared no competing interest.

infected people are 64% more likely to die than the people infected with non-resistant strains as per the WHO report [4].

CDC report of 2019 on antibiotic-resistance threats in the United States, showed that 2.8 million antibiotic-resistant infections reported every year, due to which is resulted in 35,000 deaths per year and a huge economic loss to the country [6]. European Centre For Disease Prevention and Control (ECDC) report on the bacterial challenge: "time to react", estimated about 25,000 people who have died due to antibiotic resistance problem in Europe, Iceland, and Norway, during 2007. Moreover, infection due to the antibiotic-resistant bacteria was responsible for staying patients in the hospital, extra 2.5 million days in Europe, Iceland and Norway, and these extra days may cost more than EUR 900 million [7].

There is a need to search novel effective lead molecules for future drug candidates against the antimicrobial-resistant microorganisms. In view of the above-described situation, we have developed a manually curated database of antimicrobial molecules (ACD), comprises of both synthetic chemical compounds and the antimicrobial peptides. These peptides were isolated mainly from biological sources along with some synthetic peptides. There are many databases available for antimicrobial peptides like CAMP (http://www.camp.bicnirrh.res.in/) [8], APD3 (http://aps.unmc.edu/AP/main.php) [9], PhytAMP (http://phytamp.pfba-lab-tun.org/main.php) [10] and similarly, a large number of databases are available for chemical compounds like PubChem (https://pubchem.ncbi.nlm.nih.gov/) [11], DrugBank (https://www.drugbank.ca/) [12], ZINC (http://zinc.docking.org/) [13], ChemDB (http://www.chemdb.com/) [14], etc., but none of the above databases provided a single platform where a user can get access to both synthetic chemical molecules and antimicrobial peptides. The purpose of this database was to provide a single user-friendly platform where a user can get information on both the synthetic compounds as well as the antimicrobial peptides. In ACD database, only the effective antimicrobial (compounds and peptides) agents were selected for the database. It makes easier to select the putative drug candidate. Further, this database is comprehensive in nature and provides multiple ways to find out the molecule of interest. In spite of this, the user can get good information about the properties of the molecule. The contemporary databases of ACD are generally based on data, which is very vast and nonspecific in nature, which may lead to time consuming and reduced work efficiency. The synthetic compound and antimicrobial peptide on a single platform will help researchers to go for virtual screening to identify effective lead molecules in minimum time.

## Materials and methods

### Data collection

The data collection of chemical compounds was made from the available databases ZINC [13], DrugBank [12], ChemDB [14], PubChem [11], PubChem Bioassays [15], and literature search. The antimicrobial peptides were collected from various sources like extensive literature searches from PubMed and some from Uniprot (**Fig 1**). The whole data of both synthetic compounds and antimicrobial compounds are collected based on their antibacterial and antifungal activity. Of the total data collected, about 2900 antibacterial, 1200 synthetic antifungal compounds, and about 1030 active antimicrobial peptides were selected (**Fig 2**).

The data of each synthetic compound is arranged into two categories: the first category gives information about the name and identifier of the compound (compound name, target, CBD ID, Zinc ID, PubChem ID, Canonical SMILES, IUPAC Name, ISO Smiles and Bioassay) and the second category provides information about chemical and physical properties of the compound (Molecular weight, Molecular Formula, XLogP, hydrogen bond acceptor, hydrogen bond donor and Rotatable bond).

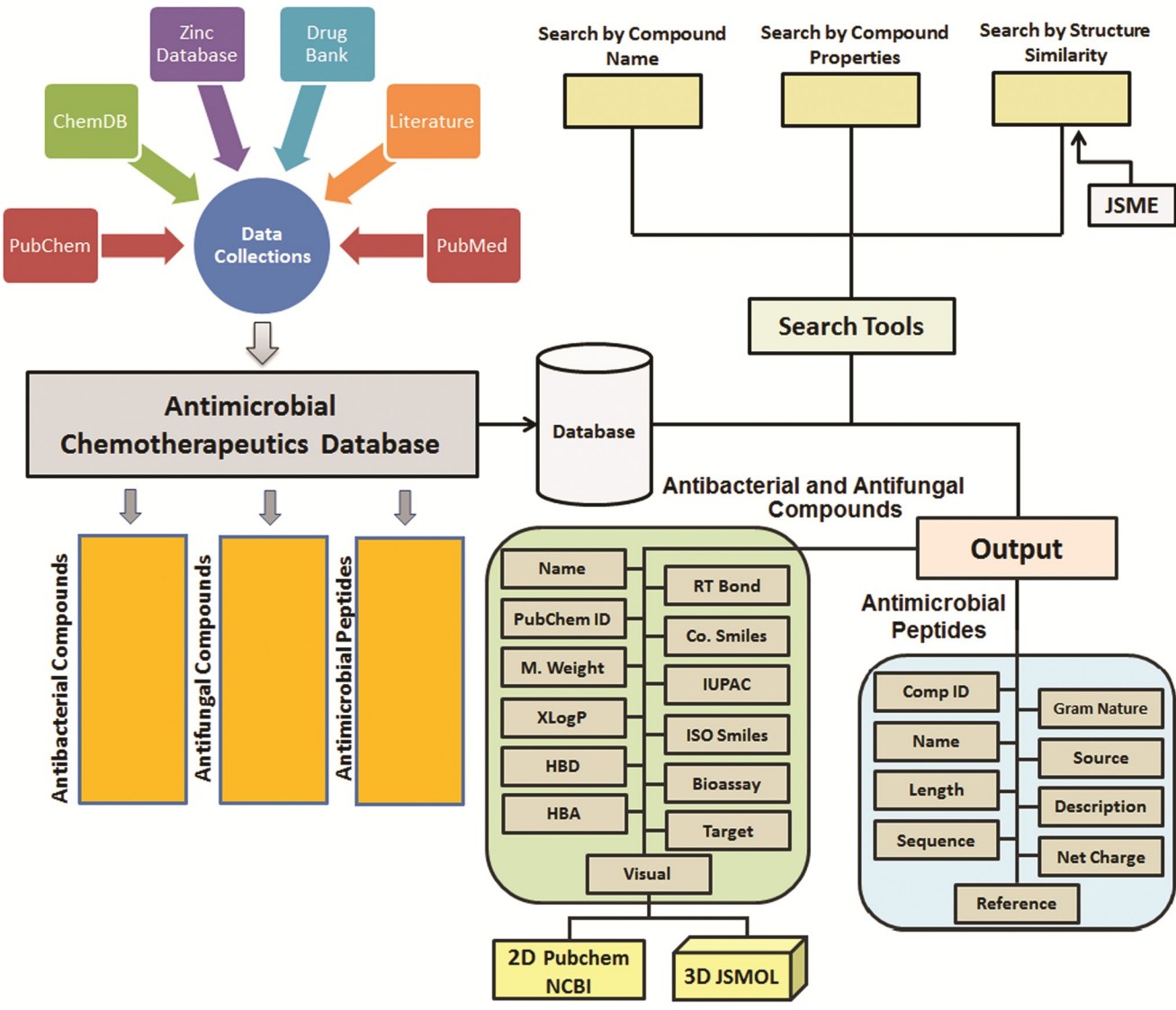

**Fig 1. Antimicrobial chemotherapeutics database architecture.**

The peptide data is categorized based on antibacterial and antifungal activity. Further, the antibacterial peptides are divided into Gram-positive and Gram-negative. The user can also get information about the peptide length, charge, target, peptide sequence, the source of the peptide, i.e. the organism from which the peptide was collected, and also a brief description of the peptide with ACD ID.

## Database framework and user interface

ACD was built on Apache Tomcat Server 7.0.76 (http://tomcat.apache.org/) over MVC Architecture with Java1.7 web technologies servlets and JSP (https://java.com/). The database tables are stored in MySQL Server 5.0 relational database. MySQL technology was preferred as they are open-source software and a platform-independent, web application developed by Java can be configured with many application servers, such as Apache Tomcat, WebLogic, JBoss, and WebSphere. Tomcat, a platform for web applications with tools for configuration and

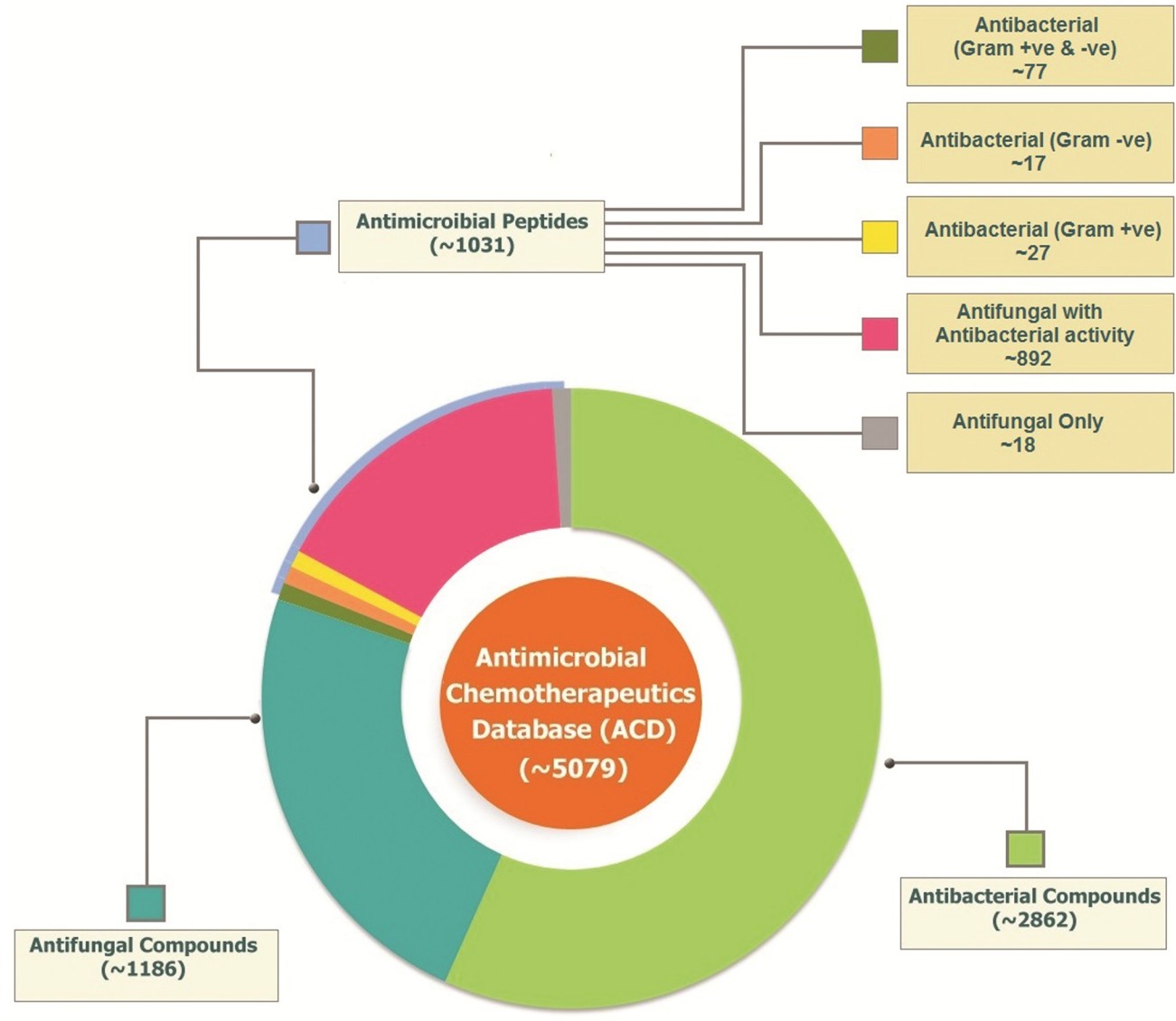

**Fig 2. Schematic representation of total compounds and peptides available in ACD.**

management, is freely available. Interfaces in the ACD database are designed in a user-friendly manner using the Bootstrap 3.0 UI Framework that allows for easy navigation over almost all kind of devices.

A brief user interface is given below:

**Home.** In this section, the various features of the ACD database are displayed through which the user can enter into its desirable section.

**Compounds.** This section deals with the synthetic antimicrobial molecules, which are categorized into antibacterial and antifungal agents. The user can search both antibacterial and antifungal compounds by their names, properties (*viz.* molecular weight, rotatable bond, H-acceptor, H-donor, and X LogP values) or by a desirable percentage of structural similarity.

**Peptide.** This database deals with the antimicrobial peptides, which can be searched by their antibacterial activity (Gram-positive or Gram-negative), antifungal activity, peptide

name or by properties of a peptide, e.g. peptide sequence, peptide length, net charge, etc. Reference and a brief description of each peptide were also provided.

**Assistance.** In this section, a user can get instructions and guide to using the various features, included in the database.

**Team.** Information is given regarding the team members involved in the database construction.

**Contact us.** Contact details of the authors are given in this section. The user can contact the authors for any query regarding the database and also can give their valuable suggestions.

## Results

### Data access

The data access in the ACD database is easy to handle. The synthetic antimicrobial compounds can be searched through antibacterial, antifungal activities, as well as by drawing structure using JSME editor and can also compress the results by specifying the percent structural similarity threshold. Each search of the synthetic compound provides information about the compound, e.g. target organisms name with its particular target if available, PubChem ID link, as well as PubChem bioassay link, are also provided so that user can visit directly to the respective databases. The user can also download the 2D SDF or 3D SDF files of the desired compound. The image of each compound was also provided.

The antimicrobial peptides were given in a separate peptide section. To make easy access to desirable peptide, we have provided several property searches, like target activity, i.e. Gram-positive, Gram-negative, antifungal, peptide length, the net charge of the peptide. The peptide sequences, a brief description of the peptide with reported MIC (mg/ml) values of most of the peptides and the references, were also given.

### Tools

**JSME.** With the help of **J**ava**S**cript **M**olecular **E**ditor (JSME), one can also draw the molecular structure of the compound to search the structural similarity [16].

**JSmol.** 3D molecular structures of the chemical compounds can be viewed through JSmol.

## Conclusion

To get-rid-of global threat of antimicrobial resistance, a need of an hour is to translate the drug research into effective medicine. High throughput screening and validation of antimicrobial agents are required to combat this global problem. Therefore, the ACD database was designed to provide a user-friendly and convenient resource for the researcher to identify novel antimicrobial agents. The ACD database provides a single platform of both antimicrobial chemical compounds and peptides. It provides several user-friendly search tools and methods, which makes the finding of the specific drug candidate much easier than other databases. A wide range of properties and structure similarity search of synthetic compounds were given so that the user can find its desirable compounds quite easily. Similarly, the effective antimicrobial peptides were selected for this database with their several properties as described earlier. The database will save time and effort of students and researchers to search for potential antimicrobial agents as future drug candidates.

## Update

The database will be updated regularly with new information and data entries from time to time. The authors and developers of this database will appreciate any comments, valuable suggestions, and queries from the users of this database.

## Acknowledgments

We would like to thank Dr. Peter Ertl for providing us Java Script Molecular Editor (JSME) [16] for our Database.

## Author Contributions

**Conceptualization:** Asad U. Khan.

**Data curation:** Mohd W. Azam, Amit Kumar.

**Resources:** Asad U. Khan.

**Software:** Amit Kumar.

**Writing – original draft:** Mohd W. Azam.

**Writing – review & editing:** Asad U. Khan.

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
