## [Decision Letter · Decision Letter 0]

19 May 2020

PONE-D-20-12977

ACD: Antimicrobial chemotherapeutics database

PLOS ONE

Dear Dr. Khan,

Thank you for submitting your manuscript to PLOS ONE. After careful consideration, we feel that it has merit but does not fully meet PLOS ONE’s publication criteria as it currently stands. Therefore, we invite you to submit a revised version of the manuscript that addresses the points raised during the review process.

We would appreciate receiving your revised manuscript by Jul 03 2020 11:59PM. To enhance the reproducibility of your results, we recommend that if applicable you deposit your laboratory protocols in protocols.io, where a protocol can be assigned its own identifier (DOI) such that it can be cited independently in the future. For instructions see: http://journals.plos.org/plosone/s/submission-guidelines#loc-laboratory-protocols

We look forward to receiving your revised manuscript.

Kind regards,

Timir Tripathi, Ph.D.

Academic Editor

PLOS ONE

Additional Editor Comments:

The database is important and the work is good but the manuscript is too short. Both reviewer feel that the manuscript may be expanded for the benefit of readers.

Journal Requirements:

" I have read the journal's policy and the authors of this manuscript have the following competing interests: [Asad U khan is editor of Plos One]"

Reviewers' comments:

Reviewer's Responses to Questions

**Comments to the Author**

1. Is the manuscript technically sound, and do the data support the conclusions?

Reviewer #1: Yes

Reviewer #2: Yes

2. Has the statistical analysis been performed appropriately and rigorously? 

Reviewer #1: N/A

Reviewer #2: Yes

3. Have the authors made all data underlying the findings in their manuscript fully available?

Reviewer #1: Yes

Reviewer #2: Yes

4. Is the manuscript presented in an intelligible fashion and written in standard English?

Reviewer #1: Yes

Reviewer #2: Yes

5. Review Comments to the Author

Reviewer #1: Manuscript Title: ACD: Antimicrobial chemotherapeutics database

Manuscript Number: PONE-D-20-12977

The ACD database is required as it clubs the properties of many important and significant databases from the prospect of antibacterial resistance.

The manuscript may be accepted with minor changes.

1.Slight syntax errors like ….”time to react ”…. there should be no space after react.; second line of second paragraph, font is different (antibiotic resistance); full form of ECDC should be given;

2. …..and very few are synthetic….This line in the abstract may be rewritten with better sentence format.

3.There should be the same format for heading. Most of them are in Capital sentences while a few are in Capital Word like Data Collection. There should be a uniformity in the entire manuscript.

4.As per 2014 WHO report…….updated data of MDR-TB should be provided. Data of 2014 is quite old for a database.

5.Tha abstract may be re-written as it is showing high similarity with the author's web page (amdr.amu.ac.in).

6.Introduction may be elaborated with more emphasis on the need of this database.

7.Comparison of available databases may be discussed in more detail. Further, compare this database with other available databases. Although the ACD database provides more information than current databases, a detail is required on the syntax of the database. Tools used to create ACD may be discussed with other databases and compare the limits and advantages of this database with other similar databases.

8.Its an excellent database so its appication and competativeness of the server should also be highlighted.

Reviewer #2: In the present database, "ACD: Antimicrobial chemotherapeutics database" author have collected huge informatios about antimicrobial compounds, and compiled it in this database. I my view it will be very useful for researcher working in the field of antimicrobial drug resistance. I have tried this database with two compounds and found positive results. I recommend this dataset for publication in PLOS ONE after two minor suggestions (If possible)

1.Add an option of search compound based on .SDF file of compound (if possible)

2.Add an option to search based on the peptide sequence (if possible)

6. PLOS authors have the option to publish the peer review history of their article (what does this mean?). If published, this will include your full peer review and any attached files.

Reviewer #1: Yes: Dr. Asimul Islam

Reviewer #2: No

---

## [Author Response · Author response to Decision Letter 0]

22 May 2020

Justification of reviewers’ comments 

Note: all changes are yellow highlighted

Reviewr 1: The ACD database is required as it clubs the properties of many important and significant databases from the prospect of antibacterial resistance. 

The manuscript may be accepted with minor changes.

1. Slight syntax errors like ….”time to react ”…. there should be no space after react.; second line of second paragraph, font is different (antibiotic resistance); full form of ECDC should be given;

Compliance: Remove space after react

"antibiotic resistance" is written in another format instead Times New Roman

ECDC: European Centre For Disease Prevention and Control 

2. …..and very few are synthetic….This line in the abstract may be rewritten with better sentence format.

Compliance: ….. as per suggestion modified “Antimicrobial chemotherapeutics database (ACD) is comprised of ~4100 synthetic antimicrobial compounds as well as ~1030 active antimicrobial peptides. The Antimicrobial peptides are mainly from biological sources and some of them are synthetic in nature.”

3. There should be the same format for heading. Most of them are in Capital sentences while a few are in Capital Word like Data 

Collection. There should be a uniformity in the entire manuscript.

Compliance:: modified as suggested by reviewr. 

4. As per 2014 WHO report…….updated data of MDR-TB should be provided. Data of 2014 is quite old for a database. 

Compliance: recent data of WHO was updated and an additional reference was also included in the text as below.

“treatment [4]. As per WHO MDR-TB factsheet 2018 report about 558,000 new cases of multi-drug resistance tuberculosis (MDR-TB) have been estimated in 2017 and 161,000 cases of MDR or RR TB detected and reported in 2017 [5]. The MRSA (Methicillin-resistant Staphylococcus aureus) infected people are 64% more likely to die than the people infected with non-resistant strains as per the WHO report [4].”

5. Tha abstract may be re-written as it is showing high similarity with the author's web page (amdr.amu.ac.in).

Compliance: abstract is modified as suggested by the reviewer

6. Introduction may be elaborated with more emphasis on the need of this database.

Compliance: Introduction is modified as suggested by the reviewer

7. Comparison of available databases may be discussed in more detail. Further, compare this database with other available databases. Although the ACD database provides more information than current databases, a detail is required on the syntax of the database. Tools used to create ACD may be discussed with other databases and compare the limits and advantages of this database with other similar databases. 

Compliance: some additional advantages of the ACD database were didcussed along with tools used to attain the objectives. 

8. Its an excellent database so its appication and competativeness of the server should also be highlighted.

Compliance: Introduction and conclusion section is modified accordingly.

Reviewer #2: In the present database, "ACD: Antimicrobial chemotherapeutics database" author have collected huge informatios about antimicrobial compounds, and compiled it in this database. I my view it will be very useful for researcher working in the field of antimicrobial drug resistance. I have tried this database with two compounds and found positive results. I recommend this dataset for publication in PLOS ONE after two minor suggestions (If possible)

1.Add an option of search compound based on .SDF file of compound (if possible)

2.Add an option to search based on the peptide sequence (if possible)

Compliance: I agree with the wornderful suggestion of reviewer but due to lock down no lab is allowed to open, students are not allowed to visit labs. It may take some time to add these featrures. But that may be added even after the study publishd. any way we keep updateing the daatbase. I assure reviewer that these features will be added later. Any way data base will remain active and functional to suffise the need of resaerchers.

---

## [Decision Letter · Decision Letter 1]

11 Jun 2020

ACD: Antimicrobial chemotherapeutics database

PONE-D-20-12977R1

Dear Dr. Khan,

We’re pleased to inform you that your manuscript has been judged scientifically suitable for publication and will be formally accepted for publication once it meets all outstanding technical requirements.

Kind regards,

Timir Tripathi, Ph.D.

Academic Editor

PLOS ONE

Additional Editor Comments (optional):

Both the reviewers are satisfied with the revisions made and suggested the acceptance of the manuscript.

Reviewers' comments:

Reviewer's Responses to Questions

**Comments to the Author**

1. If the authors have adequately addressed your comments raised in a previous round of review and you feel that this manuscript is now acceptable for publication, you may indicate that here to bypass the “Comments to the Author” section, enter your conflict of interest statement in the “Confidential to Editor” section, and submit your "Accept" recommendation.

Reviewer #1: All comments have been addressed

Reviewer #2: All comments have been addressed

2. Is the manuscript technically sound, and do the data support the conclusions?

Reviewer #1: Yes

Reviewer #2: Yes

3. Has the statistical analysis been performed appropriately and rigorously? 

Reviewer #1: Yes

Reviewer #2: N/A

4. Have the authors made all data underlying the findings in their manuscript fully available?

Reviewer #1: Yes

Reviewer #2: Yes

5. Is the manuscript presented in an intelligible fashion and written in standard English?

Reviewer #1: Yes

Reviewer #2: Yes

6. Review Comments to the Author

Reviewer #1: All the queries are answered and all the suggestions are incorporated in the revised manuscript. It may be accepted for publication in the present form.

Reviewer #2: Revised MS may be accepted for publication on PLOS ONE as it is important for the researcher working in the field of drug resistance.

7. PLOS authors have the option to publish the peer review history of their article (what does this mean?). If published, this will include your full peer review and any attached files.

Reviewer #1: Yes: Dr. Asimul Islam

Reviewer #2: No

---

## [Editor Report · Acceptance letter]

16 Jun 2020

PONE-D-20-12977R1 

ACD: Antimicrobial chemotherapeutics database 

Dear Dr. Khan:

I'm pleased to inform you that your manuscript has been deemed suitable for publication in PLOS ONE. Congratulations! Your manuscript is now with our production department. 

Kind regards, 

on behalf of

Dr. Timir Tripathi 

Academic Editor

PLOS ONE